# Photography Trilateration Indoor Localization with Image Sensor Communication

**DOI:** 10.3390/s19153290

**Published:** 2019-07-26

**Authors:** Nam Tuan Le, Yeong Min Jang

**Affiliations:** Department of Electronics Engineering, Kookmin University, Seoul 02707, Korea

**Keywords:** indoor localization, distance estimation, image sensor communications, optical camera communication, visible light communication

## Abstract

Localization has become an important aspect in a wide range of mobile services with the integration of the Internet of things and service on demand. Numerous mechanisms have been proposed for localization, most of which are based on the estimation of distances. Depending on the channel modeling, each mechanism has its advantages and limitations on deployment, exhibiting different performances in terms of error rates and implementation. With the development of technology, these limitations are rapidly overcome with hybrid systems and enhancement schemes. The successful approach depends on the achievement of a low error rate and its controllability by the integration of deployed products. In this study, we propose and analyze a new distance estimation technique employing photography and image sensor communications, also named optical camera communications (OCC). It represents one of the most important steps in the implemented trilateration localization scheme with real architectures and conditions of deployment which is the second our contribution for this article. With the advantages of the image sensor hardware integration in smart mobile devices, this technology has great potential in localization-based optical wireless communication

## 1. Introduction

In the last few decades, localization techniques that obtain the user’s location in a particular environment have been widely considered as a core technology for on-demand applications and services. This technology mainly supports industrial business solutions and includes robotic, sensor network, or human life actions such as navigation and tracking [1,2]. One of the well-known localization technologies is the Global Positioning System (GPS), which cross-references received signals from several satellites [3]. However, GPS faces many difficulties with lines-of-sight signal transmissions due to the building block propagation or signal clock. For indoor environments, this technique can be employed by re-using its simple existing infrastructure. These approaches use an additional outdoor antenna to amplify and communicate the satellite signal to an indoor antenna [4]. Various algorithms are applied for enhanced location calculation. However, these algorithms are computationally expensive and perform with low accuracy. In practice, GPS is not the optimal technique for indoor localization [5]. The development of on-demand localization services, applied both indoors and outdoors, has been an important task in wireless communication. Most localization techniques perform in two phases: distance detection and localization calculation with reference coordinators. The main factor affecting the cost and error performance of a localization technique is the distance estimation phase. When choosing calculation methods, the classification is mainly based on the signal measurement such as angle of arrival (AOA), time of arrival (ToA), received signal strength indication (RSSI), time of flight (TOF), or time difference of arrival (TDoA) [2,4,6,7,8,9,10]. Another is the signaling model base, where the classification is based on diverse designed technologies, such as RADAR, Wi-Fi, Bluetooth, UWB, visible light communications (VLC), or cellular for both indoor and outdoor scenarios.

As a new challenge in wireless communication, VLC demonstrated great achievement in its performance and in business development [11]. VLC can function at low rates, the high rate for both indoor and outdoor topologies. The main limitation of VLC is the hardware extension of the photodiode, which is a challenge for mobile integration systems, such as the smartphone. A new generation device, optical camera communication uses an image sensor from the camera for receiving, and it has an advanced hardware structure and configuration. In this study, a new approach for distance estimation and localization is proposed using OCC and photogrammetry. The proposed architecture takes advantage of the hardware on most commercial mobile devices to decode the reference coordinate. Subsequently, the position is estimated using photogrammetry on the captured image of the reference node. The article includes five sections for a detailed description of the architecture, calculations, processing steps, the state of the art with the overview of OCC, and a survey of distance estimation and localization mechanisms. Section 3 presents the architecture of the OCC system for proposed distance and localization estimation. This includes transmitter design with the communication protocol, receiver operation, and calculation process. Section 4 presents the evaluation of the performance along with the system analysis and experimental implementation. The conclusion and discussion are presented in Section 5.

## 2. State of the Art

Location information is the key component of many navigation and position services in indoor and outdoor topologies. The technical solutions for localization are well known, with different directions of approaches including active sensing (laser, IR, camera, ultrasonic, etc.) and passive sensing methods such as Wi-Fi, Bluetooth, UWB, and GPS. Table 1 summarizes the localization performance by different channel modeling. The location can be calculated using at least three distances from three reference points applying the triangle theorem. Therefore, the localization technique accuracies will depend on the accurate measurement of the distance from the device to the reference nodes. Many techniques have been proposed to address this issue, including angle of arrival, time of arrival, TDoA, RSSI, TOF, and structured light. The approach on the distance measurement based on OCC and photogrammetry is one of the promising technologies following the development of LED, image sensing, and OCC.

### 2.1. Optical Camera Communication

OCC is a new technology employed for communication following the development of VLC, which takes advantage of its optical channel, bandwidth, and security. In addition to the traditional functions, which are related to multimedia data, the communication purpose creates new challenges for the camera. As a great achievement of semiconductor and nanotechnology, the development of the image sensor in embedded systems is on the rise, and applications using the camera will be a major business trend in the near future. The camera-based communication technique has a long history, and is mainly applied using screen display topology, in particular, in vehicle communication systems in intelligent transportation. Following the development of smartphones, VLC using image sensors is a new form of optical wireless communication. It has been considered for standardization in IEEE 802.15.7m [12] with the amendment of photodiode based VLC. The current technology of the camera has some limitations regarding the frame rate, which are reflected on the data rate. However, with the development of semiconductors for new image sensor architecture and MIMO (multiple-input and multiple-output) topology, this issue will not pose a large problem for OCC in the near future. There are already some cameras currently in operation that can operate at hundreds frame per second. However, they are applied only for certain specific applications. The detailed architecture of OCC for both deployment modes is presented in Figure 1. It has the same operational concept as VLC, as the transmitter of the OCC is an LED, LED array, or a screen display device. The receiver is the camera application from the smartphone, webcam, or digital camera.

The decoding operation comprises image processing functions of the captured color pixel information. Depending on the image sensor architecture, this could be in the form of spatial or temporal data from the captured image, depending on whether the global or rolling shutter image sensor is used. With the global shutter image sensor, the entire image is captured at once. This architecture has the advantage of spatial modulation with the MIMO scenario. In contrast, the rolling shutter image sensor mode includes multiple captured states at different times in a single captured image. This architecture has the advantage of an improved data rate compared with the global shutter mode. The OCC receiving signal in this architecture is color pixel rolls, which arise from the ON–OFF light signal of the optical transmitter. By applying image processing algorithms for color classification, the receiver can decode the embedded data following the color or pixel pattern. This can be in the form of color cell status in the global shutter image sensor or frequency separation from image rolls in the rolling shutter image sensor.

One of the well-known international specifications for VLC on PHY and MAC layer from the modulation technique and error correction to medium access control function is IEEE 802.15.7. This extends to IEEE 802.15.7 m with the issues on image sensor communications. The modulation techniques in this specification include OOK (On-Off keying), FSOOK (frequency shift On-Off keying), and screen modulation. These modulation mechanisms can operate with the global shutter and the rolling image sensor. The undersampled frequency shift OOK, namely, UFSOOK [13], applies two frequencies on the OOK light signal for data modulation and one frequency for the start frame synchronization. This modulation technique is based on the binary frequency shift subcarrier. The frequencies of OOK for encoding 1 and 0 bits are separated by a specific frequency distance. The sampling frequencies of 1- and 0-bit encoding are lower than the camera frame rate for the global shutter image sensor. With two frequencies for embedded data, the proposed scheme can modulate 1 bit per sample. For synchronization, more than one frequency other than the data bit is required. It is performed with a synchronization period at each data frame. The synchronization bits operate at high-frequency OOK (10 kHz), which is above the frequency response of the image sensor. This is defined by a resulting average light intensity value from the image sensor. The main advantage of the frequency shift OOK modulation is the high-embedding bit rate per sample of the OCC system. One option is to overcome the limitation of the data rate. UFSOOK can likewise operate with the rolling shutter image at low rates. For higher data rates, the modulation can be performed by OOK, which modulates the binary bit by the ON–OFF light source. This technique has a limitation with respect to distance variation due to the captured FOV in the image sensor.

### 2.2. Overview of Distance Estimation Techniques

#### 2.2.1. Structured Light

This technique uses a single image sensor from the camera, which constructs the projected light pattern. Typically, it includes two components: the laser beams and the camera for laser triangulation. The laser beam projects a dot at a specified coordinate onto the image sensor. The position and depth of the object surface is determined by observation of the light patterns and calculation.

#### 2.2.2. AOA

The AOA technique estimates the distance using angles of arrival of signals from reference nodes. After obtaining the AOA, the position of the mobile device is determined as the intersection of multiple bearings. AOA has an advantage with respect to synchronization, because there is no requirement regarding the clock between the reference node and the mobile device. However, it has the disadvantage of complexity and cost, with the scan technique or sensor array for the LOS detection being based on the received power distribution.

#### 2.2.3. TOA

Time of arrival is the core technology for the GPS system. This technique estimates the separation distance based on the signal arrival time, which is indicated by the stamp time. This one-way propagation time is calculated by the speed of light and the carrier frequency. With the high accuracy of the time clock, TOA represents a challenge to the technology for distance estimation, because the error is 300 m at 1 μs variance. To apply this technique to location services in 2-D topology, at least three reference base station distances must be obtained.

#### 2.2.4. ToF

The concept of ToF is related to the travel time measurement of the light signal from the sensor to object target, which is similar to the ToA technique. This is the round trip time of a modulated (waveforms or pulses subcarrier) light signal. Because it is necessary to measure at the speed of light, the response of the light-sensing hardware must be sufficiently fast, namely, in the nanosecond range. The decoding process takes several duty cycles. The separation distance is calculated based on light speed, signal duty cycle, and angular frequency, which is defined by the traveling time of the signal in modulated ToF or the proportion of pulse delay in pulsed ToF.

#### 2.2.5. TDoA

The measurement is established by the time difference of the signal arriving between transmitters that simultaneously broadcast the reference signal. The TDOA technique determines the position of the mobile device based on multiple reference nodes with the distance variation. Similar to TOA, TDOA requires strict control of the signal propagation clock time, and additional communication hardware or time protocol is required to provide synchronization among the base stations and mobile devices.

#### 2.2.6. RSSI

Distance estimation can be defined by an RSS measurement, knowing the attenuation characteristics of transmitted signals. The distance between the transmitter and receiver is estimated by the signal strength, which is reduced during traveling due to the propagation effect. This is a major mechanism for distance estimation in the development and research of wireless communication. The relation between transmission, receiving power, and link distance is shown in the channel modeling ratio, the distance between transmitter and receiver, the transmission power, and the receiving power.

#### 2.2.7. Vision Photogrammetry

The main mechanism behind vision photogrammetry is based on image geometry, perspective projection, and planar holography obtained by the mono or stereo camera. Using image perspective information of the reference object and intrinsic characteristics of the camera, the distance between the camera and the object can be calculated by the initial object surface area or moving distance between two captured images. The stereo camera can provide accurate distance calculations between the camera and objects, containing information about the baseline, focal length, and corresponding disparity pixels of the left and right captured images.

### 2.3. Survey of Distance Estimation Research

Precise distance information is one of the most important requirements for driving assistance systems in the ITS system or localization for on-demand services. Most research regarding distance estimation is applied using RSSI techniques, due to their low cost and complexity. A study presented in Ref. [14] employs the TOA mechanism on UWB for high accuracy in a single base station positioning system by inverting the relations between the arriving signal amplitude and angle, which are obtained between the tag and the base stations. The distance is obtained by analyzing the flight time of the arrived signal based on the communication protocol. The ranging proposal from Ref. [7], which is based on a hybrid of TOA and RSS measurements, performs a theoretical analysis on cases of practical interest for wireless sensor network applications. Here, the inter-node separation overlaps with the region around the critical distance. The hybrid TOA/RSS maximum likelihood estimator is obtained by maximizing the joint pdf of the RSS and TOA observations. The enhanced RSS-based distance estimator [15,16] uses an iterative form of Newton’s method for maximum likelihood-based distance estimation due to the rain attenuation effect, achieving up to 90% error reduction rate. The proposed mechanism in Ref. [17] uses the Monte Carlo algorithm with variance determination by the visible light RSS-based. The calculation is built on the maximization of the signal-to-noise-ratio by means of matched filtering with intensive analytical elaborations. Likewise, working on this issue, schemes from Refs. [18,19] adopt the RSS between a base and a mobile station to obtain accuracy with particle filtering under mixed indoor LTE-A network line-of-sight and non-line-of-sight topologies. Employing another technique, the distance estimation from Ref. [20] utilizes a real-time monocular-vision-based mechanism for a vehicle scenario. The inter-vehicle distance estimation is defined by Haar-like features for vehicle detection, tail-light segmentation, and virtual symmetry detection. The detection process is based on a multi-feature fusion technique to enhance its accuracy and robustness. The distance is defined by the pixel information of the captured image with a reference angle and the distance of the camera position to the head/rear of the vehicle. This mechanism is expected to operate both day and night, and likewise for short-and long-range distances. Moreover, with the use of a single camera, the research from Ref. [21] determines the disparity map and the distance of the object from the camera system using two convex mirrors with a sufficiently long radius of curvature. The pair of images capturing the same scene from two different viewpoints is equivalent to two conventional cameras used in the stereo vision system. The study presented in Ref. [22] obtains distance based on the vanishing point, which is specified by the intersection points of straight lane lines in a vehicle scenario. Applying the Hough transform, the right and left edges are extracted and connected to the vanishing point to segment the road area and distance estimates. The solution using the infrastructure concept from Ref. [23], which represents an imaging mechanism with a blob-guided reference point, combines the pattern recognition of line laser image shapes at a certain angle. The estimation is based on the blobs–gaps relation of the position of the laser projector and the camera coordinates. The principle of the proposed scheme from Ref. [24] detects the width of the license plate as known information, and subsequently calculates the pixel size of the license plate appearing in the CCD image. From the distance–pixel relation between CCD images with known vehicle distances, the distance information can be obtained by a simple mathematical operation. Another study on the vehicle-to-vehicle topology presented in Ref. [25] uses a stereo camera for vehicle LED tracking and distance estimation. The calculation is the combination of a physical model and the Kalman filter for outdoor traveling on-road scenarios. The real-time approach for traveled distance estimation is based on a double integral of acceleration data using the built-in sensors on a smartphone during the cart movement [26]. A precise and robust distance measurement using frequency-domain analysis based on a stereo camera from the analysis of captured images was conducted in Ref. [27]. Here, distance information is calculated using the baseline of two monocular cameras, the focal length, and the pixel disparity that can be calculated directly through the phase of the frequency domain.

The distances between two LEDs in a real-world plane, AB, with the coordinate (x, y, z), and in the image plane, I_1_I_2_, with coordinate (x’, y’), are given by Equations (1) and (2).
(1)$$\mathrm{AB}=\sqrt{{\left(x_{2}-x_{1}\right)}^{2}+{\left(y_{2}-y_{1}\right)}^{2}+{\left(z_{2}-z_{1}\right)}^{2}}$$
(2)$$I_{1}I_{2}=\sqrt{{\left({x\prime }_{2}-{x\prime }_{1}\right)}^{2}+{\left({y\prime }_{2}-{y\prime }_{1}\right)}^{2}}$$

Using the pinhole camera approximation, the cross-height relation between the object and image is presented by Equation (3).
(3)$$\frac{\mathrm{AB}}{I_{1}I_{2}}=\frac{d}{f}$$

Therefore, the distance between the camera and LEDs plane, d, is given as Equation (4).
(4)$$d=\frac{\mathrm{AB}\times f}{I_{1}I_{2}},$$
where f is the camera focal length.

### 2.4. Survey of VLC Localization Estimation

The research and implementation of localization has a long history involving different levels of accuracy and complexity. VLC localization techniques can be classified into proximity, fingerprinting, triangulation, and photography vision. In the proximity technique [28,29,30], the relative position is estimated on the basis of the unique identification code of the light source, which defines one specific coordinate in the database. The accuracy level depends on the field of view of the transmitter. The mobile node can retrieve reference location offline or online using wireless communication interfaces such as Wi-Fi, Bluetooth, or ZigBee. Although this technique has low accuracy, it is facile to implement and can be applied in proximity to localization applications. The medical tracking system used in hospitals, presented in Ref. [28], estimates the location of moving devices based on the nearest ID from the illuminating LED base station. The coordinate retrieval and mapping process between mobile devices and the database server is based on the ZigBee or Wi-Fi connection interface. The enhancement of the proximity method is considered with multi-reference light sources in the presence of LED combinations in Ref. [30]. The optical element was deployed with the full angular diversity illumination from multiple LEDs by shaping their emission patterns. A simple binary signal detection of LED transmitters was proposed to identify the overlapping regions and bound the receiver location. The intersection of the optical cone lighting sources bounds the receiver location to a specific pattern. The optical signal received from a single photodiode of the LED array defines the pattern of the overlapping region by the FFT of the modulated light signal, which has different frequencies ranging from 4 to 10 kHz. The accuracy depends on the density of LEDs and the FOV of the captured signal. The fingerprinting method is a pattern recognition technique. It operates by matching the measured data with pre-measured reference location data based on the distribution of LEDs and the variance of received power due to reflections and scattering of the light source. The estimated coordinate is calculated on the basis of the linear static data or using an AI training model such as a neural network. In comparison with the proximity, the fingerprinting technique is more complex and has a higher accuracy of estimation. The proposed system can achieve average distance error is 33.5 cm and the standard deviation is 16.6 cm with a single luminary, 12.9 cm with a deviation of 8.5 cm with 9 lights, and 26 cm of average error with 4 lights.

Another technique that can achieve a higher accuracy level using the triangles’ geometric properties is triangulation. The core calculation of this technique is based on distance estimation that includes the TOA, the time difference of arrival (TDOA), the RSS, and angulation estimation with the AOA method. The VLC positioning system from Ref. [31] uses the Cramer–Rao bound for TOA-based distance estimation. The LEDs are separated by different frequencies and perfectly synchronized with the PD receiver. The VLC TDOA-based indoor positioning system from Refs. [32,33] estimates the indoor localization with PD and multiple known LEDs. The LEDs’ separation for the reference node identification is based on the TDMA technique with a square pulse pilot signal. The pilot signal can also be a sinusoidal waveform, as in Ref. [34]. The position of the mobile node was estimated from the ID detection and signal collection from PD. The proposed system from Ref. [35] uses the PD array to detect the AOA of the source signal by comparison of the received power weighted sum of angles of PDs in the PD array, which is controlled by the distance, radiance angle, and incidence angle. The result of ToA-based 3D localization experiments is obtained the precision of 100 mm to 200 mm, which would be improved by compensating the phase estimation for time synchronization from Ref. [33] and 30 cm of AOA estimation schemes based on circular PD from Ref. [35]. Working with ADOAs, the VLC positioning framework from Ref. [36] defined the angle between reference LEDs and receiver by image sensors or PD array. The scheme can achieve 15 cm error with the least-squares method. The system from Ref. [37] works with RSSI and survey data for indoor localization. The signal strength of each LED transmitter was obtained from the photo diode receiver module. The approximate position of the mobile device which holds the receiver was estimated by comparing the received values with the captured dataset value. The system includes four transmitters with a 16-cell labels arrangement for RSSI survey data.

With photography vision, the geometric relation between the world coordinates and the image plane is obtained by lens transfer using image processing. The relative coordinates of the mobile node can be derived with a single camera using co-linearity, or a dual camera using geometric distance estimation. The simulation and implementation from Ref. [38] is based on distance estimation of four preferred LEDs. The position of the mobile device is calculated by the trilateration method. The enhancement of accuracy can be considered by augmenting image sensor resolution and improving the object detection method. The localization scheme from Ref. [39] converts the pixel coordinate system of the image sensor into a mesh coordinate system using collinearity equation. The indoor localization scheme based on the OCC and PDR systems in Ref. [40] combined the identification information of the transmitter and its orientation vector to obtain an approximate location of the mobile device. The former of the rotation vector can be obtained from accelerometers and gravity data, and the latter from magnetometer and gyroscope. This is a real-time localization with a recalculated scheme to update the position and direction information of the LED. The error rate varies from 0.2 to 1.0 m, with an average value of 35 cm. Another indoor localization proposal in Ref. [41] designed an LED-based beaconing infrastructure which could be integrated to the existing lighting infrastructure. The system estimates the unknown location based on the detection image and the a priori information on the beacon positions. Each LED blinks at a high frequency to avoid flickering. Localization can be determined on the basis of LED classification using beacon and distance estimation based on LED blob size. The measurement results with 4 beacons had an error rate of 17 cm.

## 3. Proposed Localization Estimation Scheme

### 3.1. Methodology

#### 3.1.1. Distance Estimation

The proposed scenario of distance estimation based on the image sensor is shown in Figure 2. The scheme uses at least two LEDs, which continuously broadcast their coordinate information using VLC technology. From the captured coordinate information on the image sensor, the distance between the two LEDs is given by Equation (5).
(5)$${\mathrm{AB}}^{2}={\left(X_{A}-X_{B}\right)}^{2}+{\left(Y_{A}-Y\right)}^{2}+{\left(Z_{A}-Z_{B}\right)}^{2},$$
where X, Y, and Z are coordinates of the LEDs in the real world, which are represented with A and B; f is the camera focal length. In addition, I_1_I_2_ is the distance between the two LEDs’ image in the image sensor with the x’ and y’ coordinates, which is given by Equation (2).

The estimation process includes two main calculations: the image sensor rotation distance and the center shift distance. First, the image sensor rotation distance defines the proportion distance between the two LEDs and their image in the image sensor by means of a rotation angle. It includes two processing steps, which define the image of LEDs on the OXY plane with a separation of the Z coordinator of the LEDs and the image sensor plane with a rotation angle. If the camera image sensor is parallel to the LED plane, as in Figure 2, the distance, calculated using Equation (4), is the proportion among the focal lens, the reference LEDs, and the pixel length of the image of the LEDs. However, with a general captured case, such as that shown in the scenario illustrations in Figure 3a, 3b and 3c, where the captured image sensor plane is not parallel to the LED plane, the distances between the LEDs on the image sensor are the same. For the general scenario, the calculation must consider the captured angle that comes from the lines between the two LEDs and the camera surface.

As shown in Figure 3d, with the assumption that A and B are known LEDs, the image of AB on the OXY plane is AB’, the image of AB on the image sensor is I_A_I_B_, the distance from AB to the image sensor can be defined by pinhole camera approximation, and is d. However, the desired value is h, which can be calculated as follows:(6)$$\{\begin{array}{c} {\mathrm{AB}}^{\prime }=\mathrm{AB}\times \cos\left(µ\right)\\ d=\frac{{\mathrm{AB}}^{\prime }}{I_{B}I_{A}}f\\ \mathrm{AD}=\frac{d}{f}I_{A}I_{F}\\ {\mathrm{AF}}^{2}={\mathrm{AD}}^{2}+d^{2}\\ \mathrm{Cos}\left(\Phi \right)=\frac{d}{\mathrm{AF}}\\ h=\cos\left(µ-\Phi \right)\times \mathrm{AF} \end{array}$$
where µ is angle between AB and AB’, µ is also equal to the angle between HF and FD. Then, ɸ is the angle between AF and FD. Finally, I_F_ is the center point of the image sensor.

Considering the image sensor rotation status in Figure 4, the rotation angle with OXY plane is controlled by the pitch and roll axis information. These can be acquired from the variation detected by the Gyroscope Sensor with a reference coordinate or direction coordinate of the two reference LEDs. The calculation for distance estimation with the scenario in Figure 3d assumes that the image sensor plane and the OXY plane are parallel. For a general rotation of the image sensor to the OXY plane, the calculation of µ in Equation (6), which directly affects AB’ (as Figure 3d), is redefined by µ_oxy_ as in Figure 5, where µ_oxy_ is the angle between AB and the OXY plane; µ_OXY_ is the angle between AB and the OXY plane. The rotary image sensor plane has a normal vector, which is defined in Equation (7).
(7)$$\vec{n}=\left[\vec{u}\times \vec{v}\right],$$
where $\vec{u}\;\mathrm{and}\;\vec{v}$ are the rotation vectors of the x-world coordinate axis and the y-world coordinate axis, respectively, given by Equation (8). β and ∝ are the rotation angle of image sensor on Roll axis and Pitch axis.
$$\vec{u}=(\cos\left(\propto \right),0,\sin\left(\propto \right))$$
$$\vec{v}=(0,\cos\left(\beta \right),\sin\left(\beta \right))$$
(8)$$\vec{n}=\left(-\sin\left(\propto \right)\cos\left(\beta \right),-\cos\left(\propto \right)\sin\left(\beta \right),\cos\left(\propto \right)\cos\left(\beta \right)\right)$$

The angle between the LED line and the camera image sensor plane in Equation (5) is as follows:(9)$$µ=\vec{\mathrm{AB}}\cap \left(\mathrm{oxy}\right)$$
(10)$$\sin\left(µ\right)=\frac{\left|\vec{\mathrm{AB}}.{\vec{n}}_{\mathrm{oxy}}\right|}{\left|\vec{\mathrm{AB}}\right|\times \left|{\vec{n}}_{\mathrm{oxy}}\right|}$$
(11)$$\sin\left(µ\right)=\frac{\left|X_{\vec{\mathrm{AB}}}\times X_{{\vec{n}}_{\mathrm{oxy}}}+Y_{\vec{\mathrm{AB}}}\times Y_{{\vec{n}}_{\mathrm{oxy}}}+Z_{\vec{\mathrm{AB}}}\times Z_{{\vec{n}}_{\mathrm{oxy}}}\right|}{\sqrt{X_{\vec{\mathrm{AB}}}{}^{2}+Y_{\vec{\mathrm{AB}}}{}^{2}+Z_{\vec{\mathrm{AB}}}{}^{2}}\times \sqrt{X_{{\vec{n}}_{\mathrm{oxy}}}{}^{2}+Y_{{\vec{n}}_{\mathrm{oxy}}}{}^{2}+Z_{{\vec{n}}_{\mathrm{oxy}}}{}^{2}}},$$
where $\vec{\mathrm{AB}}$ = (X_B_-X_A_,Y_B_-Y_A_,Z_B_-Z_A_).

The second calculation is for the center shift distance, which defines the real distance between the plane of the two LEDs and the image sensor with a shift distance from the center of the image sensor, as shown in Figure 6. The expected distance, EF, is defined by Equation (12).
(12)$$\{\begin{array}{c} {\mathrm{EF}}^{2}={\mathrm{EG}}^{2}+{\mathrm{GF}}^{2}\\ \mathrm{GF}=\frac{\mathrm{CD}\times f}{I_{C}I_{D}}\\ \mathrm{EG}=\frac{I_{G}I_{E}\times \mathrm{GF}}{f} \end{array}$$

Here, I_G_ is the center point of the image sensor, which is generated from the principal axis of a lens and the image sensor. FG, which depicts the distance between the LED plane and the lens in the center of the image sensor, can be calculated from the first step. Subsequently, the I_G_I_E_ is the center shift of the LED image, which can be measured from image sensor pixels through the perpendicular line of the image of the two LEDs and the image sensor center point.

The equation of the line from the two points (I_A_, I_B_) with the slope, m, and the intercept of the line, b, is given by the following conditions (13).
(13)$$\{\begin{array}{c} y=\mathrm{mx}+b\\ m=\frac{Y_{I_{A}}-Y_{I_{B}}}{X_{I_{A}}-X_{I_{B}}}\\ b=Y_{I_{A}}-mX_{I_{A}} \end{array}$$

In the image sensor plane (oxyz), the coordinates of I_E_ are defined by the intersection point between I_A_I_B_ and the perpendicular line to I_A_I_B_ traveling through the image sensor center point, given by Equation (14).
(14)$$\{\begin{array}{c} y=\mathrm{nx}+c\\ m\times n=-1\\ c=Y_{I_{G}}-mX_{I_{G}} \end{array}$$

Then, the coordinates of the intersection point, E, from the intersection between the expectation distance line and the line between the two LEDs in the LED plane are calculated by Equation (15).
(15)$$\{\begin{array}{c} \mathrm{AE}=\frac{I_{A}I_{E}\times \mathrm{EF}}{{\mathrm{FI}}_{E}}\\ y=\mathrm{kx}+d\\ k=\frac{Y_{A}-Y_{B}}{X_{A}-X_{B}}\\ d=Y_{A}-kX_{A} \end{array}$$
where I_A_I_E_ is calculated based on the coordination of detected I_A_ in the image sensor and defined by the formula in Equation (14) for point I_E_.

#### 3.1.2. Coordinate Estimation

In the general scenario shown in Figure 6, the intersection point, E, between the LED line, AB, and the distance of the line between LEDs plane to camera, EF, from the distance estimation step, includes the complete coordinates based on trilateration formulas. On the basis of this mechanism, every pair of transmitter LEDs can define one distance between the camera and the LED pair at the specified intersection point. With n LEDs, we have k combinations of n sets of distance (between the camera and the LED line) and full coordinate of intersection point.

In the general scenario, the localization of mobile devices can be estimated on the basis of reference nodes with known coordinates and measured distances. With n known anchor nodes and distances from the mobile node to anchor nodes, L_i_ = (x_i_,y_i_), r_i_, where (i = 1, …, n), the estimated location of mobile node M = (x,y) in two dimensions can be calculated by Equation (16) [42]. The anchor nodes and distance are defined in Section 3.1.1, especially Equations (12) and (15).
(16)$$\{\begin{array}{c} {\left(x_{1}-x\right)}^{2}+{\left(y_{1}-y\right)}^{2}=r_{1}{}^{2}\\ {\left(x_{2}-x\right)}^{2}+{\left(y_{2}-y\right)}^{2}=r_{2}{}^{2}\\ \ldots \ldots .\\ {\left(x_{n}-x\right)}^{2}+{\left(y_{n}-y\right)}^{2}=r_{n}{}^{2} \end{array}$$

The calculation can be represented in matrix form as Ax = b, which defines the relationships between positions and distances according to Equation (17).
$$A=\left[\begin{array}{c} 2\left(x_{n}-x_{1}\right)\;2\left(y_{n}-y_{1}\right)\\ 2\left(x_{n}-x_{2}\right)\;2\left(y_{n}-y_{2}\right)\\ \ldots \ldots .\\ 2\left(x_{n}-x_{n-1}\right)\;2\left(y_{n}-y_{n-1}\right) \end{array}\right]$$
(17)$$b=\left[\begin{array}{c} r_{1}{}^{2}-r_{n}{}^{2}-x_{1}{}^{2}-y_{1}{}^{2}+x_{n}{}^{2}+y_{n}{}^{2}\\ r_{2}{}^{2}-r_{n}{}^{2}-x_{2}{}^{2}-y_{2}{}^{2}+x_{n}{}^{2}+y_{n}{}^{2}\\ \ldots \ldots .\\ r_{n-1}{}^{2}-r_{n}{}^{2}-x_{n-1}{}^{2}-y_{n-1}{}^{2}+x_{n}{}^{2}+y_{n}{}^{2} \end{array}\right]$$

The position of the mobile node with coordinates (x,y) can be calculated using the least-squares matrix by Equation (18).

x = (A^T^A)^−1^A^T^b(18)

It can be seen that the accuracy of the coordinate calculation is mainly controlled by the distance estimation results. However, due to the Gaussian distributions, and inaccurate anchor positions and distance measurements, the least-squares formula can be reassigned with a weight of estimation variances as Equation (19). By applying more reference nodes which Gaussian distribution error for the distance estimation, the error with respect to the coordinates can be reduced.
$$A=\left[\begin{array}{c} 2\left(x_{n}-x_{1}\right)\times w_{1}\;2\left(y_{n}-y_{1}\right)\times w_{1}\\ 2\left(x_{n}-x_{2}\right)\times w_{2}\;2\left(y_{n}-y_{2}\right)\times w_{2}\\ \ldots \ldots .\\ 2\left(x_{n}-x_{n-1}\right)\times w_{n-1}\;2\left(y_{n}-y_{n-1}\right)\times w_{n-1} \end{array}\right]$$
(19)$$b=\left[\begin{array}{c} (r_{1}{}^{2}-r_{n}{}^{2}-x_{1}{}^{2}-y_{1}{}^{2}+x_{n}{}^{2}+y_{n}{}^{2})\times w_{1}\\ \left(r_{2}{}^{2}-r_{n}{}^{2}-x_{2}{}^{2}-y_{2}{}^{2}+x_{n}{}^{2}+y_{n}{}^{2}\right)\times w_{2}\\ \ldots \ldots .\\ (r_{n-1}{}^{2}-r_{n}{}^{2}-x_{n-1}{}^{2}-y_{n-1}{}^{2}+x_{n}{}^{2}+y_{n}{}^{2})\times w_{n-1} \end{array}\right]$$

Here, $w_{i}=1/\sqrt{\sigma_{{\mathrm{distance}}_{i}}^{2}+\sigma_{x_{i}}^{2}+\sigma_{y_{i}}^{2}}$.

The calculations from Equations (16) to (19) for x, y, and z correlates with the least-squares system for three reference points in a 2D topology and four reference points for a 3D topology.

### 3.2. OCC Transmitter

With the limitations of OOK modulation on the distance variation due to the captured FOV of the image sensor, FSOOK demonstrates the advantages in terms of stripe width separation. At different distances of the communication link, the covered bits in the captured image will be inversely proportional to the distance. This means that the amount of covered bits at a greater communication distance will be smaller. The bit loss arises from the coverage problem of the camera FOV. This cannot be recovered by any technique due to the out of protocol control rule. This is one of the main limitations of OOK modulation for the OCC system. With the frequency subcarrier, where the modulated bit is represented by multiple cycles, the distance variation does not affect the decoding process. In this technique, the binary bit classification is defined on the basis of the waveform frequency of the subcarrier, which is represented by a group of rolling stripes at the image sensor. This demonstrates the advantage of bit encoding compared with OOK modulation. At different link distances, the subcarrier frequency can be classified by the Fourier transform or the stripe width pattern. The number of rolling stripes depends on the captured FOV of the image sensor and the link distance. However, if the width of rolling stripes is constant, then the subcarrier frequency will also be the same. This is the basic advantage of FSOOK modulation compared with the OOK technique with respect to bit decoding.

The frequency shift OOK modulation operates with multiple frequency shifts of the on-and-off LED light. The number of frequency shifts decides the number of embedded bits in one symbol. The frequency selection should consider eye safety and the threshold frequency of the shutter. Basically, these frequencies range from 100 Hz to 8 kHz for complete illumination with flickering mitigation, as shown in Figure 7. However, to reduce the down-rate of the coding bit due to the line coding mechanism, besides the frequencies for data coding, one more frequency should be allocated for the preamble signal. The preamble signal operates at the beginning of every super frame for synchronization. Most existing studies on the frequency shift OOK modulation define the preamble frequency above the image sensor shutter speed threshold. This is about 10 kHz, at which the captured image is at the average brightness level of the light source.

Flickering is the first consideration in VLC deployment, which is defined as one of the most important factors of the transmitter. The frequency separation is defined by the difference in width between the black and white stripes, which is created by the on-and-off signal duration of the LED, as shown in Figure 8. If there is a big variation in the average illumination, the flickering point will be recognized by the human eye, as shown in Figure 9. The higher the frequencies that are applied, the more flickering will occur because of the larger distance of the frequency switch. This is a tradeoff between the embedded rate and the flickering index of the flickering point.

To minimize the flickering point, the proposed system applies two frequencies for the subcarrier on–off signal. The bit encoding mechanism and frequency selection are shown in Table 2. The line coding mechanism is defined with two sampling bits “00” for the synchronization signal. At the frequencies of 2 and 4 kHz, the flickering point is minimized to 10% with variation in illumination. The system can achieve a flickering index of 90% with the “VISO system flicker tester” application [43]. The frequency selection is based on two factors: the flickering point and the communication reliability.

With regard to communication, if low frequency is selected for subcarrier, there is a tradeoff relation between the numbers and the pixel width of the LED captured roll. The number of rolls is decreased when the captured distance is increased. This is reliable for decoding the signal. However, the flickering point is large because of the illumination variation between on and off light signals at the frequencies of f1 and f2. The optical clock rate of the transmitter should be the same as the camera frame rate of the receiver. The optical rate and camera sampling rate selection should consider the image processing for the bit decoding algorithm.

### 3.3. OCC Receiver

The proposed OCC receiver architecture is based on the rolling shutter mode of the image sensor. The modulated optical signal is captured and converted to a color pixel array by the camera. The out stream is a buffer at which the output image data are dumped at each sampling frame. The data decoding includes light conversion, memory access time, and data image processing. The total time must be less than 1/(frame rate). If this value is larger than the buffer, the memory will overflow. The memory access time is dependent on the image type format and resolution. To minimize the decoding time, we applied a non-compressing image method, based on the gray-scale YUV color space. Figure 10 presents the data decoding with image processing. The gray-scale image data from the buffer are dumped to bitmap data using a two-dimensional array to minimize the processing of contour detections that result from the color conversion.

The data decoding process includes two steps: detection of the regions of interest (ROIs) and subsequent extracting cell data. Among these, the first step isolates the image areas from the captured image. Subsequently, the second step decodes the frequency from detected ROIs. The ROI separating step is more important, because it can cause a fatal error in the next step. While detection of high-frequency changes in illumination is conducted, we exploit the fact that the rolling shutter of most modern CMOS cameras does not capture total image simultaneously. Alternately, the data transfer is pipelined by the sensor with the exposure of a pixel. This means that a light pulsing at a frequency, which is much higher than the capturing time of a frame, will light up only some rows, thus producing bands in the image. By detecting the frequency of these bands in the frequency domain of the image and inferring the pulsing light frequency, we can use the relationship between them to build up the demodulation for FSOOK.

The main difference in the OCC receiver system configuration is the exposure time, which is defined as the pixel roll exposure duration. Under a different configuration of the exposure time, which follows the shutter speed, the light density is spread out onto the image sensor during absorption. This can likewise reduce the interference or the blur effect of being near the lighting source. Then it has a direct effect on the frequency separation in the decoding process. The large reduction configuration, however, has the disadvantage of low brightness, as the captured image is portrayed at a dark gray level. Therefore, the shutter speed configuration should consider the brightness of the scenario, which depends on LED power and the environment.

The on–off signal is presented by black and white stripes of occupied pixels in the captured image. The width of the stripes depends on modulation frequency. With a square waveform signal at a frequency f_s_, the duration of the on–off state, t_f_, will be 1/(2f_s_) seconds for a pair of black and white stripes. To generate one stripe in h_s_ pixel rows with readout time, the width of the stripe pair w_s_, in millimeters, is defined by Equation (20).

The blur effect, which arises from the inter-symbol interference and the offset shifting time of exposure process, has a width given by Equation (21).
(20)$$\{\begin{array}{c} t_{r}=\frac{t_{f}}{h_{s}}\\ h_{s}=\frac{t_{f}}{t_{r}}\\ w=h_{s}\times d_{c}=\frac{\frac{1}{2f_{s}}}{t_{r}}\times d_{c}=\frac{1}{2f_{s}\times t_{r}}\times d_{c} \end{array}$$

Here, d_c_ is the pixel density, t_r_ is the row read out time, and h_s_ is the height of the image in pixels, as shown in Figure 11.
(21)$$h_{\mathrm{blur}}=\frac{t}{t_{r}},$$
where t is the exposure time at which the sensor opens the shutter to receive photons.

If the transition time of the modulation optical signal is larger than the camera exposure time, the captured signal cannot be classified. The stripe width in Figure 12 shows the effect of different shutter speed configurations. At short shutter speeds, the stripe width is reduced to close to the optical clock rate.

### 3.4. Topology and Operation

The flowchart for distance estimation based on OCC is shown in Figure 13. It includes five steps: OCC transmitter classification, coordinate query, device rotation matrix calculation, OCC LED image coordinate tracking, and distance estimation. In the first step, the OCC LEDs are extracted from among the illumination LEDs to be used for data communication in the second step. The OCC LEDs broadcast their coordinates through the optical channel. The coordinates of the two LEDs are queried from a server with the unique IDs, which are obtained by the OCC technique at the third step. The query can be executed with the index table or communication technology such as Wi-Fi or LTE. The distance between two LEDs is obtained from the coordinate information. Subsequently, the separation distance of two LED images on the image sensor plane can be calculated by pixel measurement. By combining the rotation angle between the LED lines and the image sensor plane at the fourth step, distance estimation can be finalized.

### 3.5. Error Analysis

The proportions of h and a in the camera architecture shown in Figure 14 are defined by Equation (22). The accuracy of the object image height a will affect the error rate of d, where the focal length f and the object height h are defined values extracted from the camera information and OCC reference coordinate calculation.

The estimation accuracy is directly dependent on the LED image landmark recognition. More specifically, it is strongly affected by the reference point for coordinate identification, which depends on the light conditions of the environment and the reference landmarks. The specified coordinates of the proposed scheme are defined on the basis of the center of the LED contours, as shown in Figure 15. However, because of the rolling effect portrayed in Figure 16, the captured stripes of LED are not at a fixed location in the image plane. This generates error with regard to the center position of LED contours.
(22)$$\frac{h}{a}=\frac{d}{f}$$
(23)$$f_{\mathrm{pixels}}=\frac{f_{\mathrm{mm}}\times \mathrm{pixelDensity}}{25.4}$$
(24)$$d=\frac{h\times f_{\mathrm{pixels}}}{a}$$
(25)$$\partial d=\frac{h\times f_{\mathrm{pixels}}}{a\left(a-1\right)}$$

With the error of pixel amounts in the image line of the two LEDs, the distance calculation is proportional to h and f. A 1-pixel error at the image sensor plane can generate an error in the distance, according to Equations (23)–(25). With common camera parameters, e.g., f = 600 pixels, the distance is equal to ~3 cm for a 1-pixel error.

The clarity of the stripes in the captured image affects the data decoding error and distance estimation accuracy, which are based on the stripe width separation. Assuming that a white LED is used as a transmitter, the width of the stripe comes from the color band between the black and white regions of the two LED off-state stripes. The most important detection parameter is the on–off separation threshold value, which is controlled by the transition between black and white and the LED driver frequency response.

The absorbed photon intensity of the LED signal at the image sensor with the LOS path loss model is given by Equation (26).
(26)$$P_{r}=\int_{0}^{T}h\left(t\right)⊗p\left(t\right))\mathrm{dt},$$
where h_i_ is the color channel DC gain, P(t) is the transmission power of the light source, and P_r_ is the received optical power.

The channel DC gain path loss can be defined with Equation (27).
(27)$$\begin{array}{c} H_{\mathrm{LOS}}\left(0\right)=\frac{\left(m+1\right)A}{2{\pi D}^{2}}\cos^{m}\left(\varphi \right)T_{s}\left(\theta \right)g\left(\theta \right)\cos\left(\theta \right),\\ 0\leq \theta \leq \theta_{\mathrm{FOV}}, \end{array}$$
where m is the order of the Lambertian emission, A is the image sensor pixel area, D is the distance between the transmitter and the receiver, Ɵ is the angle of irradiance, T_s_(Ɵ) is the angle of incidence, g(Ɵ) is the signal transmission coefficient of an optical filter, cos(Ɵ) is the gain of an optical concentrator, and Ɵ_FOV_ is the receiver field of view. The values of m and g are defined by Equation (28).
(28)$$\begin{array}{c} m=\frac{\ln2}{\ln\cos\left(\varepsilon \right)},\\ g\left(\theta \right)=\frac{n^{2}}{{\sin\theta }_{\mathrm{FOV}}{}^{2}}, \end{array}$$
where n denotes the internal refractive index of the optical concentrator.

With the effect of the camera lens, the channel gain is defined by Equation (29).
(29)$$H_{\mathrm{LOS}}\left(0\right)={\;H}_{\mathrm{LOS}}\left(0\right)\;\times {\;C}_{\mathrm{blur}},$$
where $C_{\mathrm{blur}}$ is the blur concentration ratio, which is quantified by its standard deviation by Equation (30).
(30)$$C_{\mathrm{blur}}=\frac{s^{2}}{\pi {\left(\frac{f\;\times \;l}{f}+\sigma_{\mathrm{blur}}\right)}^{2}/4},$$
where s, f, and l are the pixel edge length, camera focal length, and the diameter of the LED source.

To analyze the performance of the system’s SNR over signal propagation, the noise model plays an important role. Optical noise includes shot noise ∂, thermal noise, and multipath inter-symbol interference [29], defined by Equations (31)–(34), respectively.
(31)$$N=N_{\mathrm{shot}}+N_{\mathrm{thermal}}+N_{\mathrm{interference}},$$
(32)$$N_{\mathrm{shot}}=2\mathrm{qW}\mu \left(P_{\;r}+P_{\mathrm{rISI}}\right)B_{\mathrm{en}}+2{\mathrm{qWI}}_{\mathrm{bg}}I_{2}B_{\mathrm{en}}+\;\varepsilon ,$$
(33)$$N_{\mathrm{thernal}}=\frac{8{\pi \mathrm{KT}}_{k}}{G}{\delta \mathrm{AI}}_{2}B_{\mathrm{en}}{}^{2}+\;\frac{16\pi^{2}{\mathrm{KT}}_{\mathrm{kE}}}{g_{m}}\delta^{2}A^{2}I_{3}B_{\mathrm{en}}{}^{3},$$
(34)$$N_{\mathrm{interference}}=\mu^{2}{\mathrm{Pr}}_{\;\mathrm{rISI}}^{2},$$
where µ is the detector responsibility, q is the electronic charge, $B_{\mathrm{en}}$ is the equivalent noise bandwidth, W is the sampling rate of the camera, I_bg_ is the background current, $I_{2}$ is the noise bandwidth factor, and ε is the affection noise from the neighbor channel. K is the Boltzmann constant, T_k_ is the absolute temperature, G is the open-loop voltage gain, δ is the fixed capacitance of the photo detector per unit area, E is the FET channel noise factor, g_m_ is the FET trans-conductance, I_3_ = 0.868, and I_2_ = 0.562. $P_{r}$ denotes the received power, and $P_{\mathrm{rISI}}$ is the received power resulting from the inter-symbol interference. T depicts the exposure time of the camera.

The SNR value is defined by Equations (35) and (36).
(35)$$\mathrm{SNR}=\frac{S}{N}$$
(36)$$\begin{array}{c} S=\mu^{2}{\mathrm{Pr}}_{\;\mathrm{Signal}}^{2}\\ P_{\mathrm{rISI}}=\int_{T}^{2T}(\sum h_{i}\left(t\right)⊗P_{i}\left(x\right))\mathrm{dx} \end{array}$$

With the proposed 2FSOOK modulation at 2 and 4 kHz of the subcarrier, the BER is defined by Equation (37).
(37)$$\mathrm{BER}=Q\left(\sqrt{\mathrm{SNR}}\right)$$
Figure 17 and Figure 18 show the performance of the SNR and SINR with different configurations of shutter speed and link distance. For the SNR, the fluctuation of the signal strength is controlled by the received power propagation and the lens blur effect. In contrast to SNR, the main factor of SINR is the exposure time, which avoids the interference of the optical source from neighbor modulated light pulses.

## 4. Performance Evaluation

### 4.1. Image Sensor Communication Performance

The distance and localization estimation performance of the proposed scheme was evaluated by applying the experiment implementation circuit for the transmitter, including the microcontroller LED driver and the LED, as shown in Figure 19. The receiver comprises a smartphone rolling shutter camera and an application that can configure the camera shutter speed.

The data rate of the proposed 2FSOOK modulation scheme at different configured camera frame rates is shown in Figure 20. The system data rate is proportional to the camera frame rate. However, high camera frame rates create a problem with decoding processing. The experiment performed on the current commercial smartphone at more than 20 fps is not reliable with respect to the processing capability, due to the cache memory. The buffer is overloaded in some scenarios where the shot noise is high. Table 3 provides the configuration of the experiment that guarantees data encoding and decoding.

The roll stripes of subcarrier frequencies of 2FSOOK are shown in Figure 21, where the 2 and 4 kHz OOK frequencies are embedded. In this experiment, the implementation of a 17 cm diameter of LED and the power density distribution analysis of two neighbors LED optical pulse at 2 and 4 kHz OOK frequency are shown in Figure 22. The distance from the transmitter to the receiver can be at 2 m for reliable decoding data with identified stripes in the captured image. At a larger distance, the separation distance between the contours is not stable, and the system encounters an error resulting from missing stripe widths, which arise as a result of the rolling mechanism limitation. Therefore, to achieve high accuracy and minimize the complexity of error correction, the maximum number of rolling contours should be guaranteed for the demodulation of one frequency. The number of contours depicts the maximum communication range between the transmitter and the receiver. This means that with FSK modulation, the position of the receiver can change dynamically, whereas the maximum communication range will be recognized when the number of contours is more than one. In our implementation, we configure the image sensor with a resolution of 600 × 800 pixels and a maximum communication range of 2 m. The higher the resolution configuration, the longer the distance we can achieve for communication. However, this setup faces processing time issues on the device. The performance with regard to processing capability with respect to image formats, resolution configuration, and the rolling signal pixel width of 2FSOOK are shown in Figure 23, Figure 24 and Figure 25.

### 4.2. Distance Estimation Performance

The performance of the proposed scheme for distance estimation is measured using photography with the setup illustrated in Figure 26. As described in the analysis from the previous section, the estimation error arises from the calculation of the distance between the LEDs, which is defined by the line connecting the two center points of reference LEDs. The OCC LEDs can be separated into common LEDs by continuous rolling contours. The coordinates of the LED ROI are covered by the position of the first and last contour, which are varied at every captured frame due to the rolling effect with the on status of LED. In the experiment, Figure 27a,b represent the features of the 4 and 2 kHz LED signal, which correspond to the maximum of 4 and 6 pixels at 600 × 800-pixel resolution. The error of the coordinates of the LED contour on the image sensor ranges from 1 to 6 pixels under stable conditions.

A photograph depicting the experimental implementation of the proposed scheme is presented in Figure 28, portraying two OCC LEDs among three illumination LEDs. The coordinates of the two OCC LEDs are mapped with the index table through LED-ID [44] or by receiving the coordinates directly from the LED signal. In this scenario, we applied the LED-ID architecture, where there is 5-bits ID. The data are broadcast by the LEDs, which are 17 cm apart, and received by the rolling shutter camera in the smartphone. The coordinates of the LEDs are queried by a server via a Wi-Fi connection. With regard to the camera focal length, this is one of the most important parameters for distance estimation using the pinhole camera approximation. It should be defined initially for the estimation processing. Its value can be obtained from the Camera API (for example, CameraIntrinsics or CameraCharacteristics library for Android devices). It can also be defined initially through manual configuration using a built-in database. The focal length of existing cameras will be surveyed and adapted for the application. By extracting the device model from the camera API, which most existing devices can support, we can match the device camera focal length from the database and then configure these values.

For the centralization error rate analysis of the image sensor photography technique, the experiment implementation only considered the optical camera communications and pinhole camera approximation calculation. The camera focal length value is calibrated by manual configuration to minimize the error value. In addition, then, the rotation angle between the mobile device and the LED plane, which can be extracted from the gyroscope sensor, is also considered with the fixed scenario. The rotation angle with the gyroscope sensor was expected to be a minimum of 2 degrees in Ref. [45]. We conduct the experiment by measuring 1,000 times in order to calculate the average across mean and variance. The performance of distance estimation is shown in terms of the cumulative distribution function (CDF) results in Figure 29. It can achieve an average estimation error of 10 cm, which arises from the error of the LED center point detection due to the projected contours.

### 4.3. Localization Estimation Performance

The performance evaluation of the localization estimation is based on the scenario portrayed in Figure 30. It includes three main steps, extracted from four processes: OCC detection, coordinator download, ranging estimation, and coordinator estimation. Firstly, the mobile terminal downloads the coordinates of the three reference LEDs using OCC. Secondly, it references the LED range calculation based on photography geometry. Finally, it performs the localization estimation using trilateration. Among these four steps, OCC detection plays an important role in all system functions that support data decoding, coordinator mapping, and OCC receiver classification.

The experimental setup for localization estimation is shown in Figure 31a–c, which portray the off-status of reference LEDs, non-communicative mode of camera, and communication mode of camera, respectively. In this scenario, there are three data LEDs that coexist with other illumination lighting LEDs. In the non-communicative mode, the camera shutter speed is configured at 30 Hz, at which there is no classification between the illumination LEDs and communication LEDs. However, in the communication mode, the inner optical interference is removed by the high speed of the camera shutter. Using a computer vision algorithm, we can easily separate the OCC LEDs and illumination LEDs. As discussed in the previous section, the main consideration with respect to the OCC data rate is the processing time capacity at the receiver, which is affected by the image processing algorithm, resolution, and the number of sources.

As a result of distance estimation in Section 4.2, which is controlled by camera focal length, rotation matrix angle and LED ROI, the performance evaluation of system localization, which is defined by trilateration formulas, has another factor, Gaussian distributions error of anchor distances. The performance of 300 localization estimation measurements with three reference nodes, as in the configuration scenario in Section 4.2, is shown in Figure 32. In this experiment, the vertical value, which is defined by the “z” coordinate, is calculated on the basis of the photography image sensor calculation of Section 3. The origin of the coordinate system and the direction of the axis are defined by the received reference coordinates and the image sensor coordinates. The performance accuracy of more reference coordinates with Gaussian weighing is shown in Figure 33.

## 5. Conclusions

Considering the importance of localization services in the business trends of on-demand applications, the proposed scheme for distance estimation based on OCC and photogrammetry shows the advantages of VLC in visible light optical channels and the development of image sensors in smart devices. The proposed system includes a non-flickering visible light system transmitter, an OCC rolling shutter receiver, and employs a photogrammetry mechanism for distance estimation. The main contributions of this article are the OCC system design and implementation, and the distance estimation and localization calculation mechanism using OCC and photography. The performance results, as well as advantages of the hardware and low complexity, imply a new promising technique for indoor localization services. In future work, the performance evaluation of the camera’s focal length calibration and rotation angle of image sensor will be considered with a thorough analysis.

## Figures and Tables

**Figure 1 sensors-19-03290-f001:**
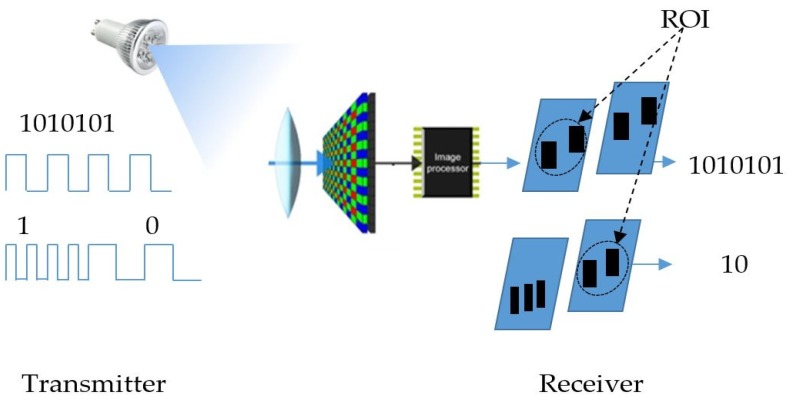
OCC architecture.

**Figure 2 sensors-19-03290-f002:**
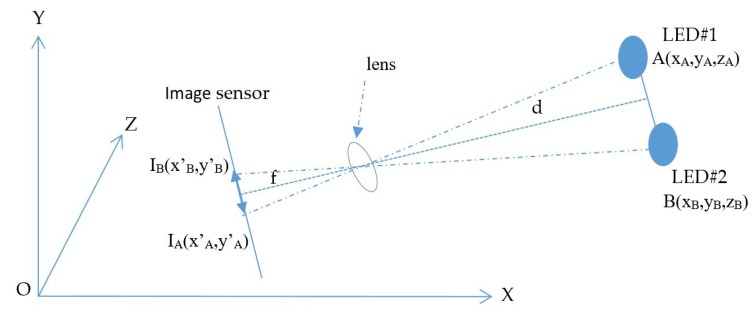
Distance-estimation-based photography technique.

**Figure 3 sensors-19-03290-f003:**
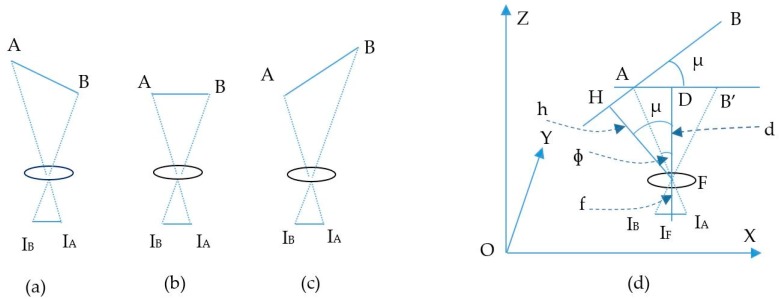
Illustration of different captured angle scenarios. (**a**), (**c**) LEDs line not parallel with image sensor plane. (**b**) LEDs line parallel with image sensor plane. (**d**) Image of LEDs line in image sensor plane.

**Figure 4 sensors-19-03290-f004:**
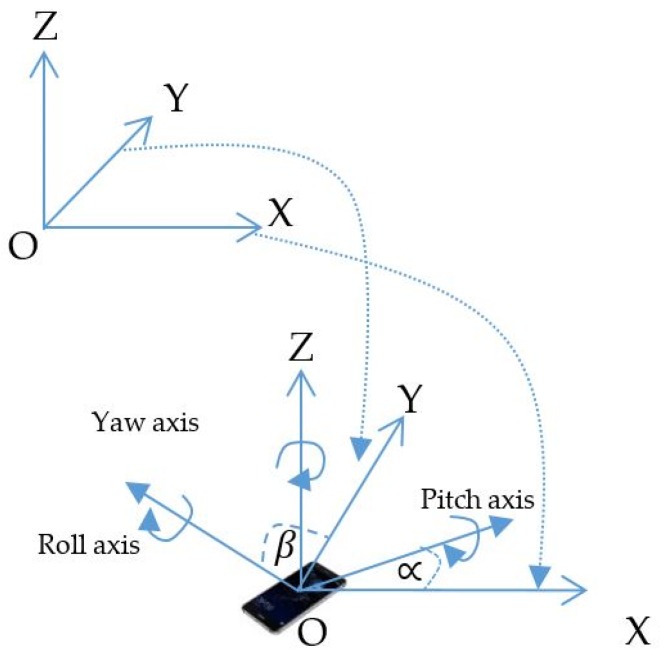
Image plane rotation scenario.

**Figure 5 sensors-19-03290-f005:**
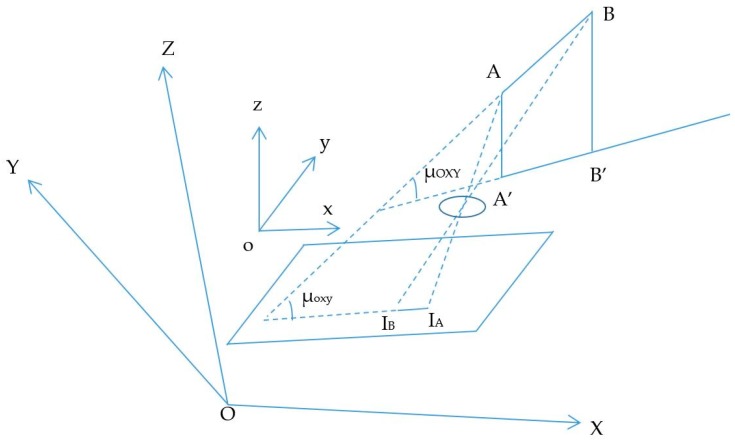
Image of reference points in the image sensor plane.

**Figure 6 sensors-19-03290-f006:**
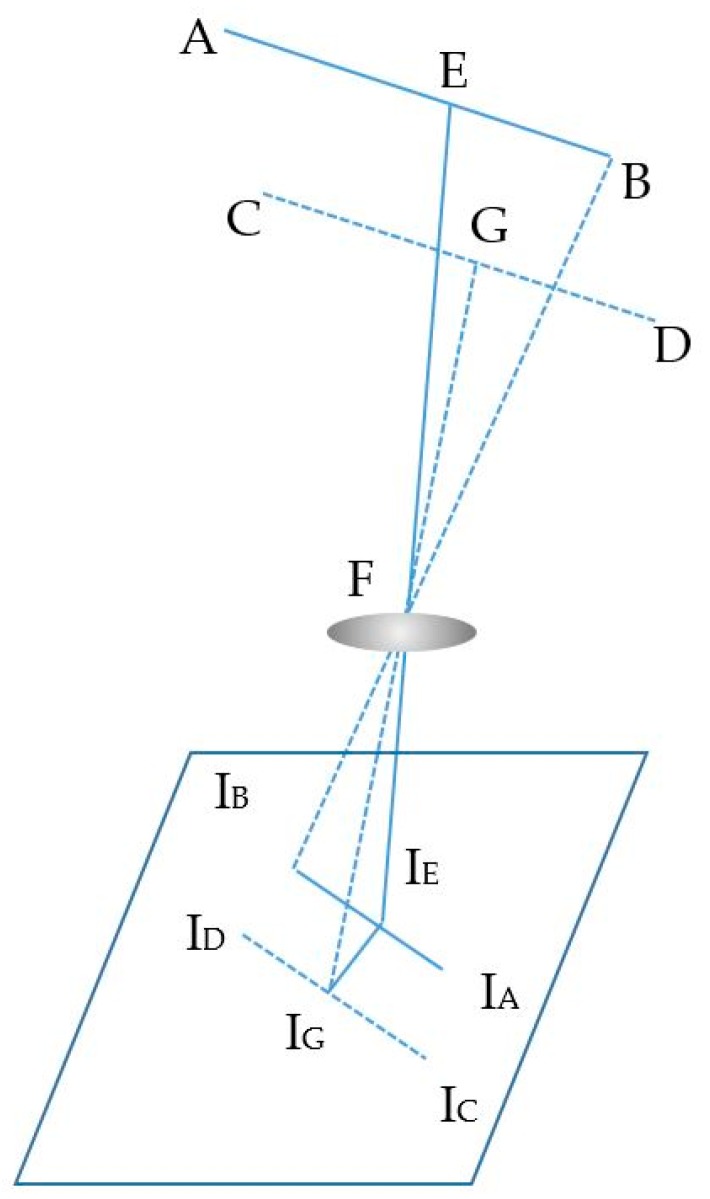
Center shift scenario.

**Figure 7 sensors-19-03290-f007:**
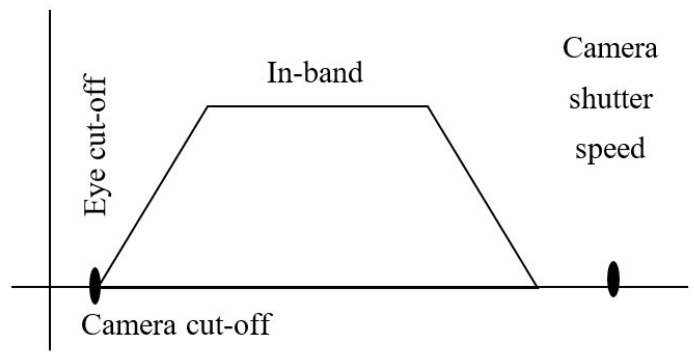
OCC frequency band.

**Figure 8 sensors-19-03290-f008:**
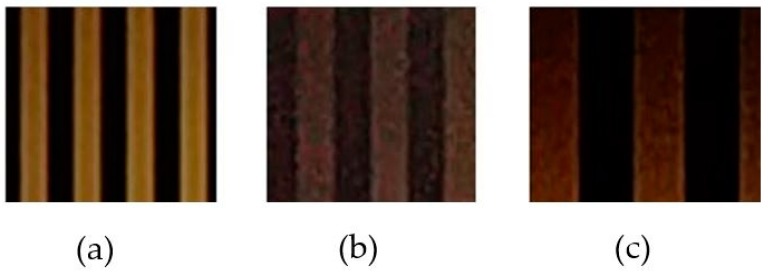
OOK signal at 2 kHz (**a**), 1 kHz (**b**), and 500 Hz (**c**).

**Figure 9 sensors-19-03290-f009:**
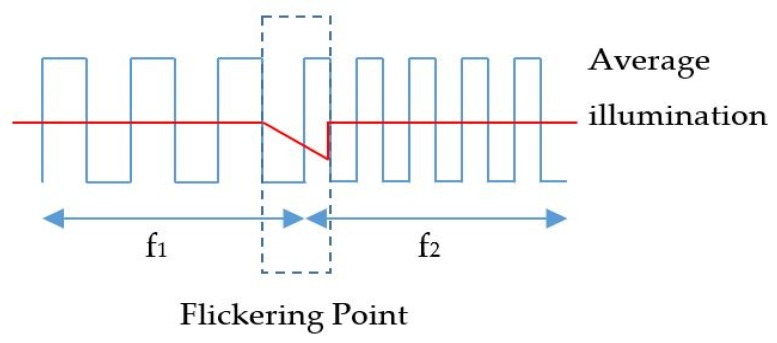
OCC flickering.

**Figure 10 sensors-19-03290-f010:**
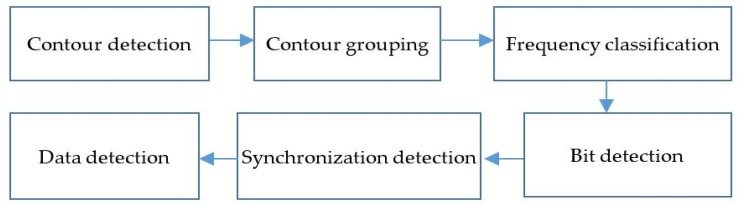
Data decoding processes.

**Figure 11 sensors-19-03290-f011:**
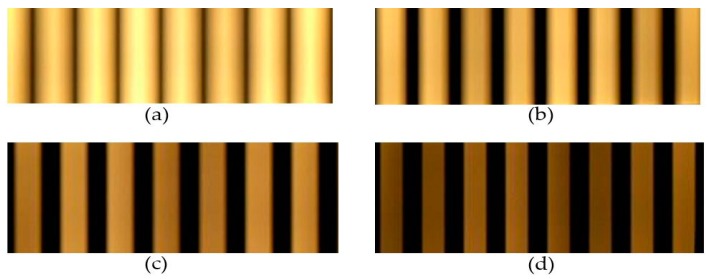
Effect of shutter speed in the received image. (**a**) 1000 Hz of shutter speed. (**b**) 2000 Hz of shutter speed. (**c**) 4000 Hz of shutter speed. (**d**) 8000 Hz of shutter speed.

**Figure 12 sensors-19-03290-f012:**
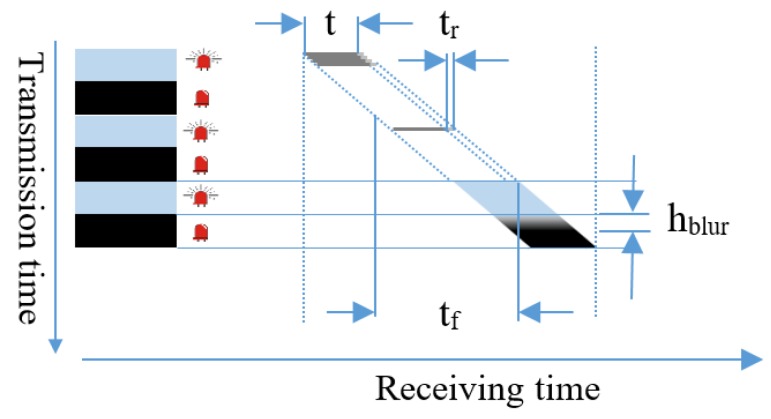
Rolling image sensor operation.

**Figure 13 sensors-19-03290-f013:**
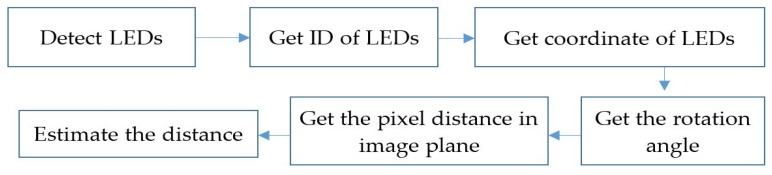
Distance estimation processes.

**Figure 14 sensors-19-03290-f014:**
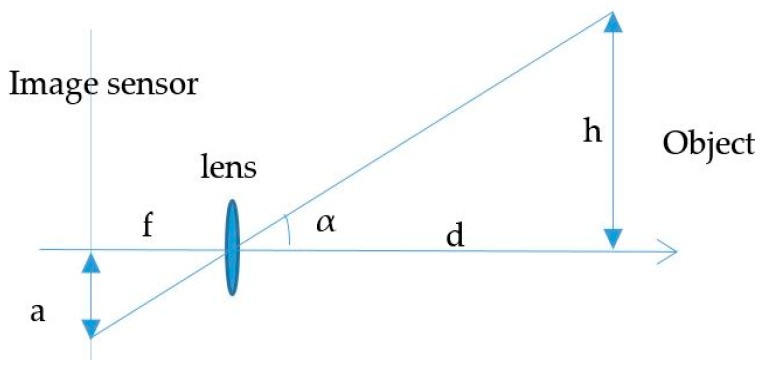
Pinhole camera.

**Figure 15 sensors-19-03290-f015:**
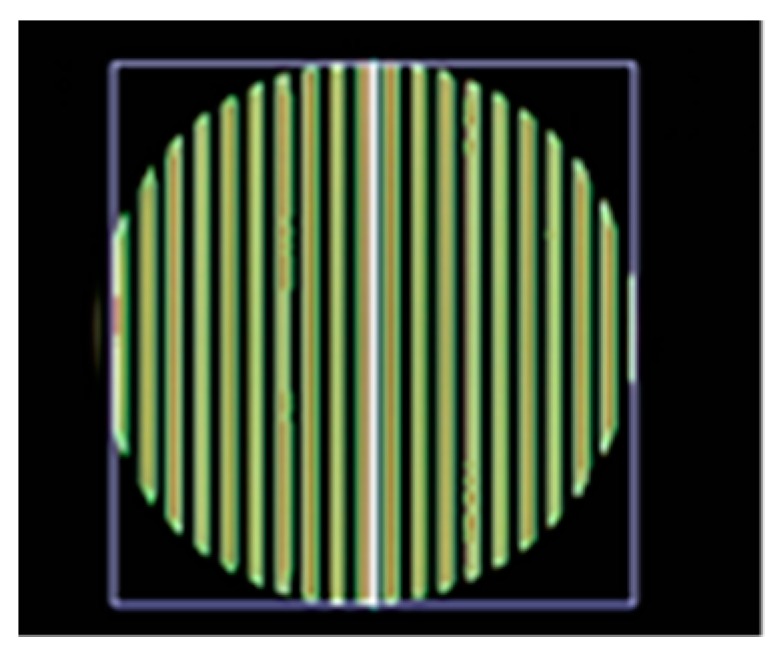
Reference coordinates on image sensor plane.

**Figure 16 sensors-19-03290-f016:**
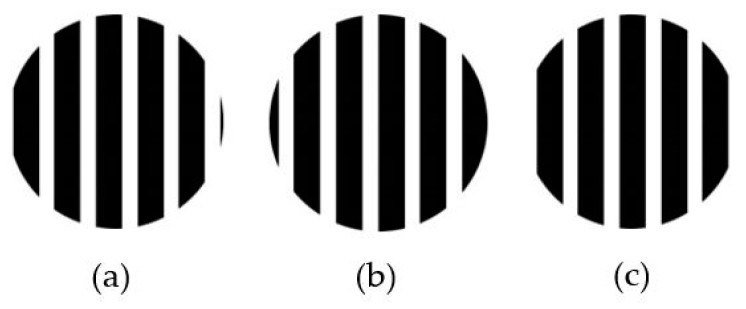
Captured contour scenarios. (**a**) Stripe starts at ON state and finishes at OFF state. (**b**) Stripe starts at OFF state. (**c**) Stripe starts at ON state and finishes at ON state.

**Figure 17 sensors-19-03290-f017:**
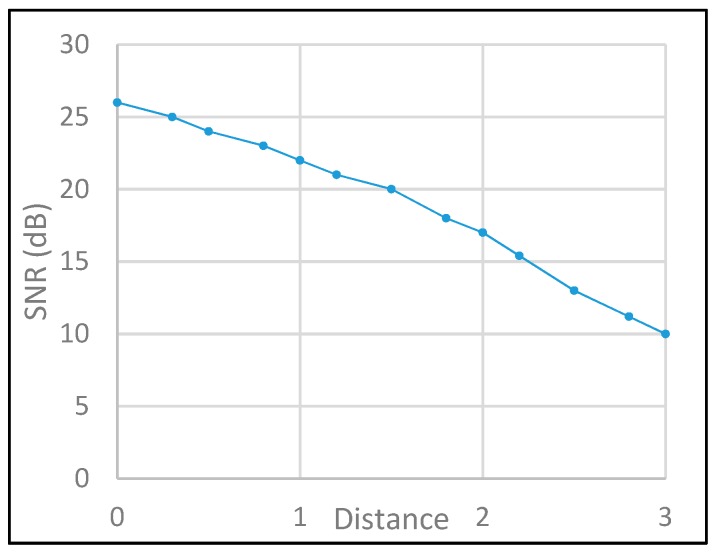
SNR evaluation.

**Figure 18 sensors-19-03290-f018:**
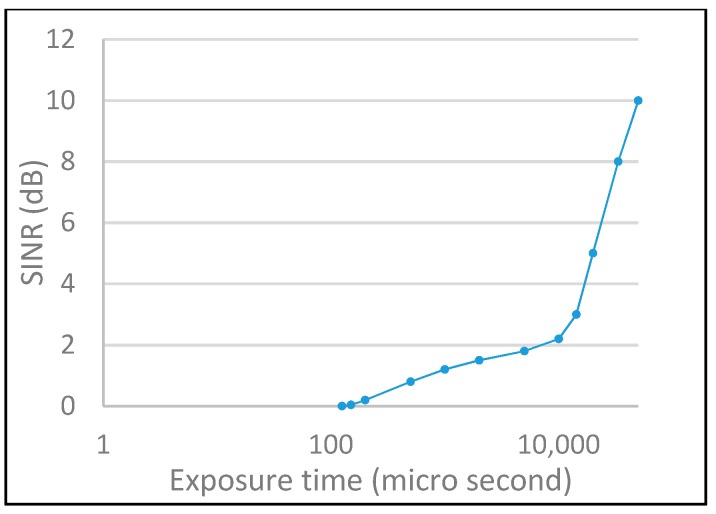
SINR evaluation.

**Figure 19 sensors-19-03290-f019:**
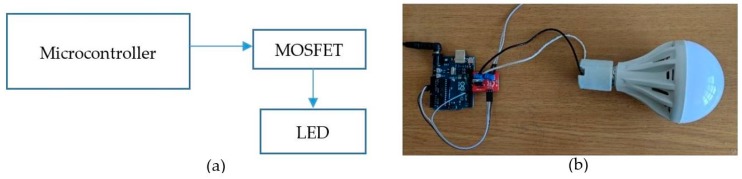
Transmitter implementation architecture. (**a**) Circuit diagram. (**b**) Devices demo.

**Figure 20 sensors-19-03290-f020:**
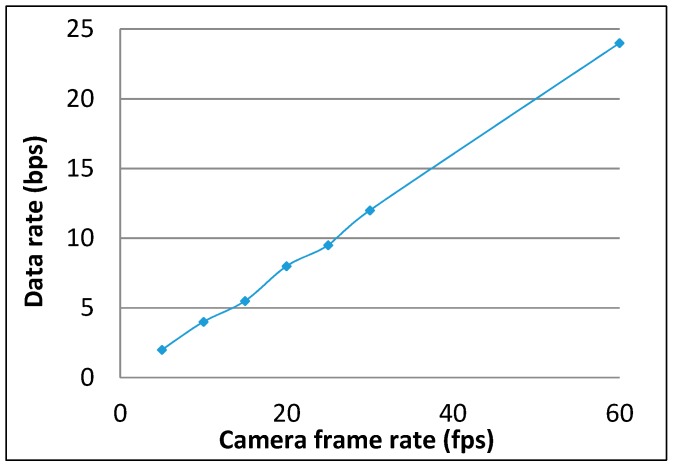
Data rate performance of 2FSOOK.

**Figure 21 sensors-19-03290-f021:**
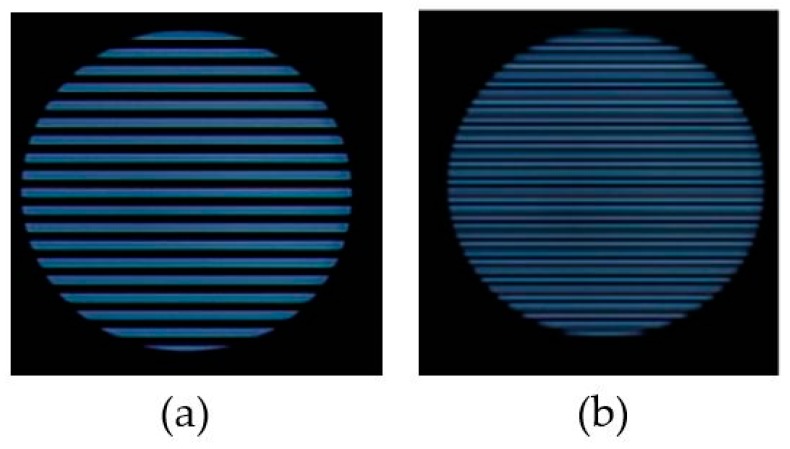
Rolling shutter image of 2FSOOK. (**a**) 2 kHz signal. (**b**) 4 kHz signal.

**Figure 22 sensors-19-03290-f022:**
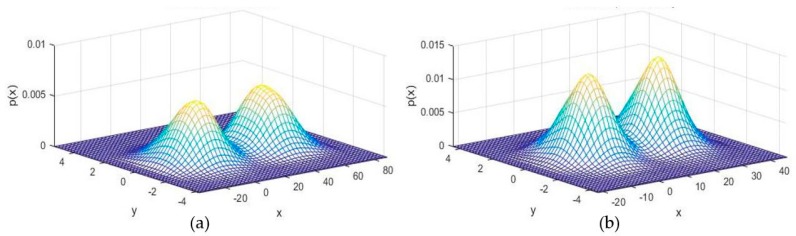
Power density of LED pulse at 2 kHz OOK subcarrier (**a**) and 4 kHz OOK subcarrier (**b**).

**Figure 23 sensors-19-03290-f023:**
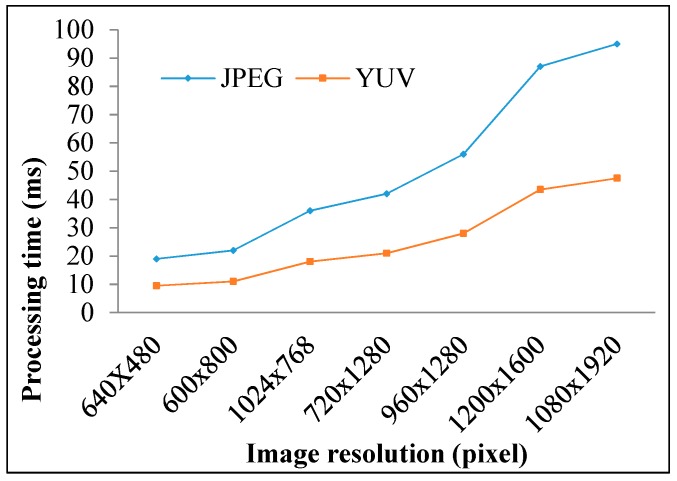
Processing performance vs. image sensor resolution.

**Figure 24 sensors-19-03290-f024:**
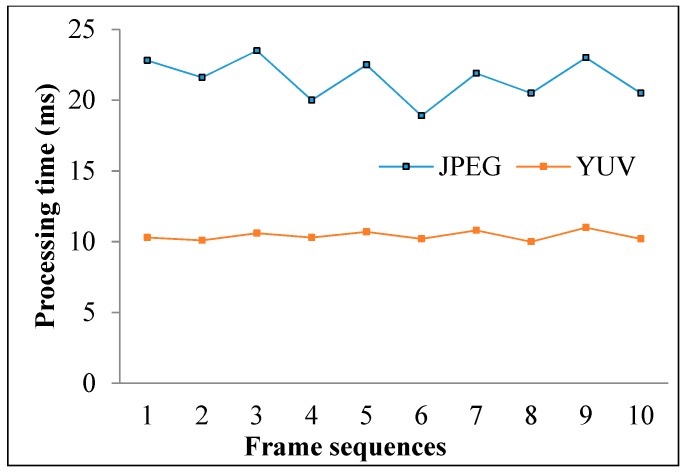
Processing performance vs. image type configuration.

**Figure 25 sensors-19-03290-f025:**
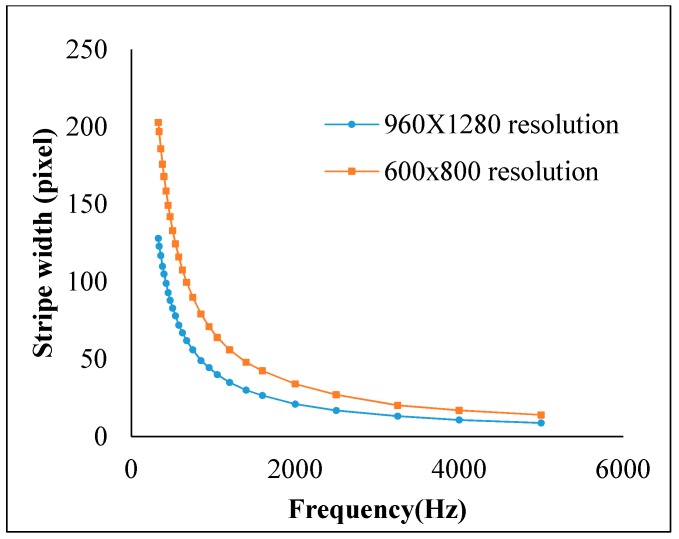
Roll width feature with respect to signal frequency and image sensor resolution.

**Figure 26 sensors-19-03290-f026:**
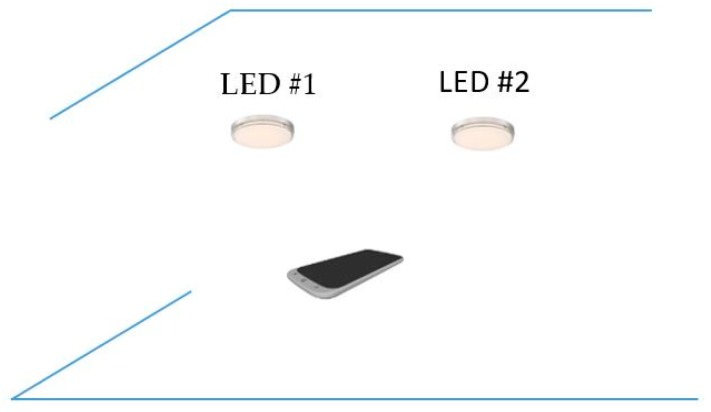
Distance estimation scenario.

**Figure 27 sensors-19-03290-f027:**
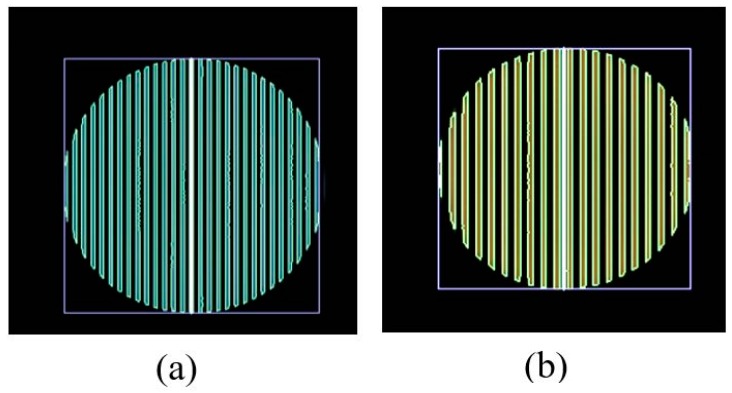
ROI contour feature. (**a**) Captured image of 4 kHz signal. (**b**) Captured image of 2 kHz signal.

**Figure 28 sensors-19-03290-f028:**
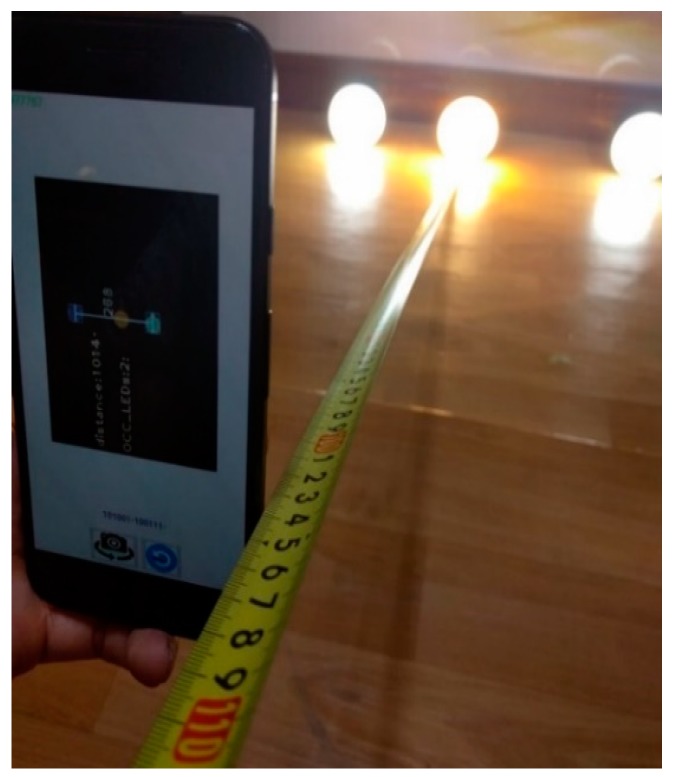
Distance estimation experiment.

**Figure 29 sensors-19-03290-f029:**
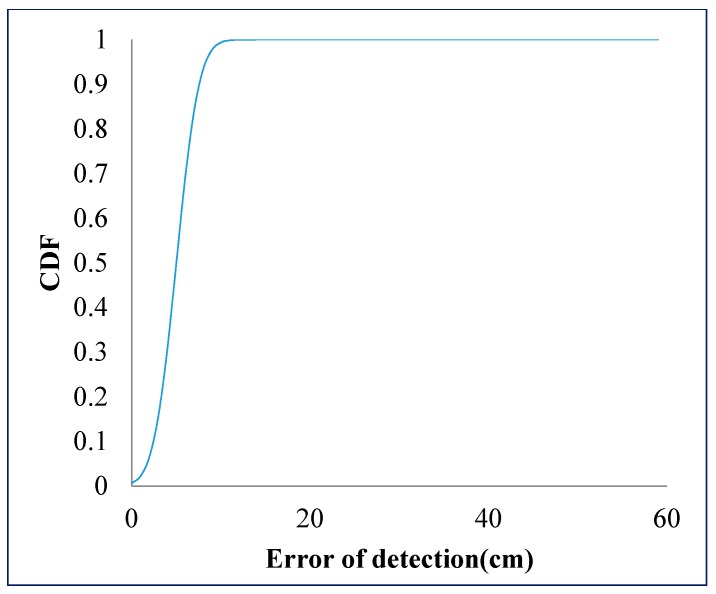
CDF of distance error performance.

**Figure 30 sensors-19-03290-f030:**
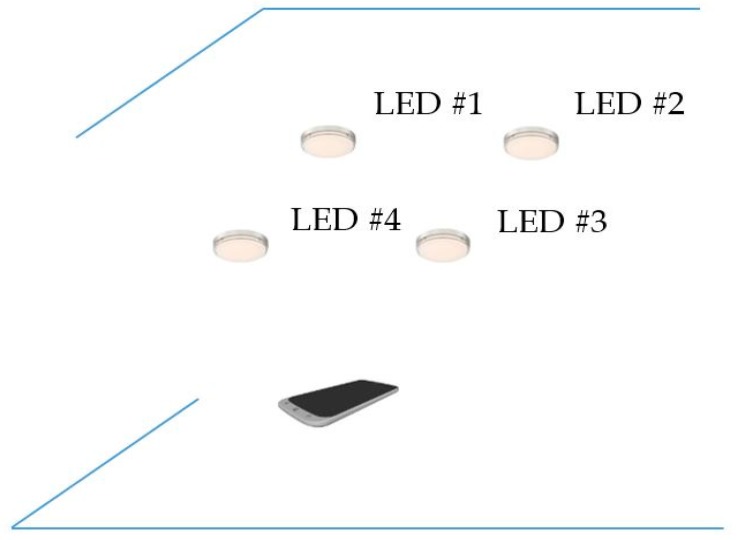
Localization estimation scenario.

**Figure 31 sensors-19-03290-f031:**
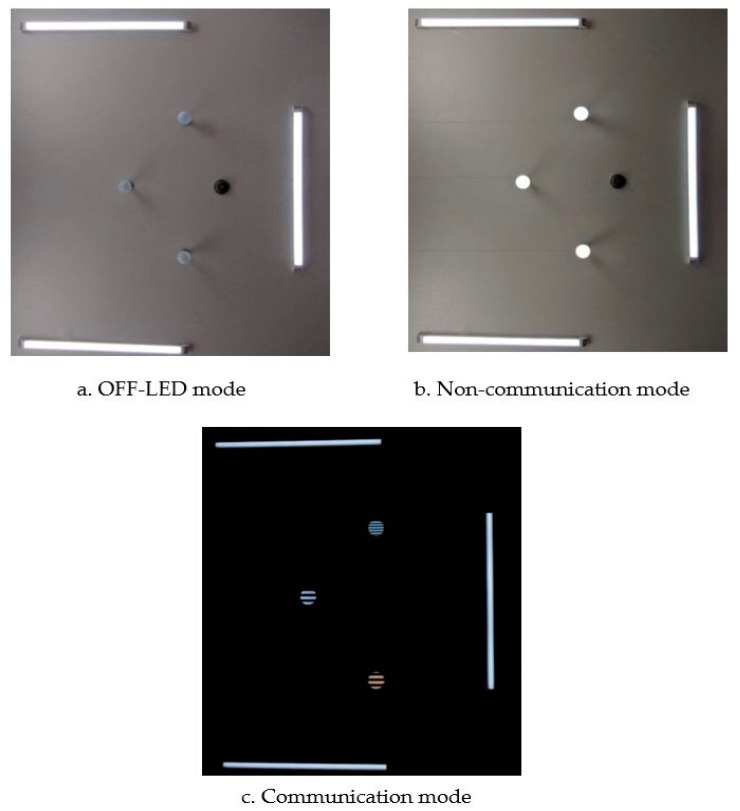
Evaluation experiment scenario. (**a**) Off state of LEDs scenario. (**b**) Illumination state of LEDs scenario. (**c**) Communication state of LEDs scenario.

**Figure 32 sensors-19-03290-f032:**
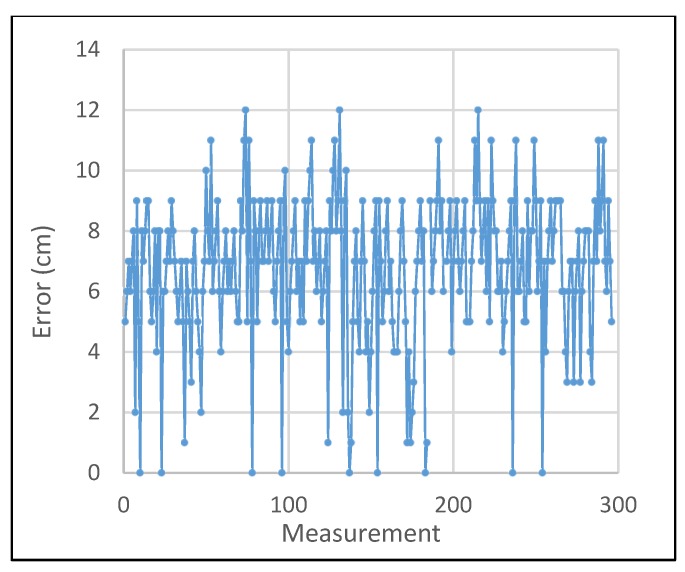
Localization error performance.

**Figure 33 sensors-19-03290-f033:**
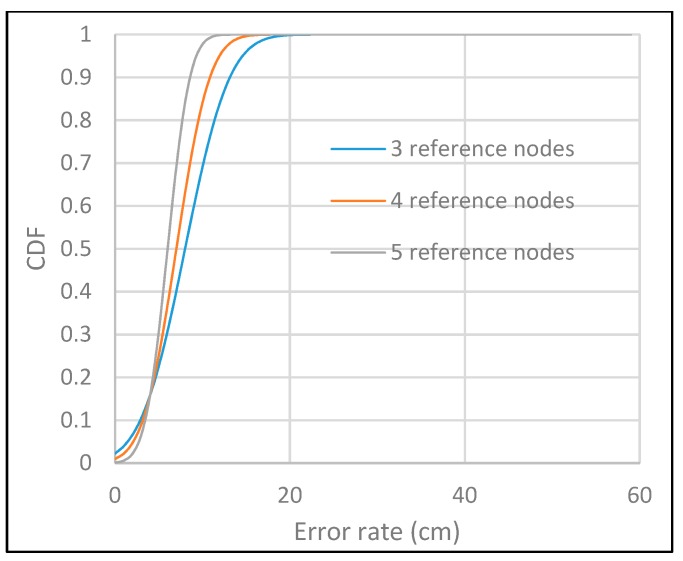
CDF performance of localization.

**Table 1 sensors-19-03290-t001:** Summary of localization performances.

Technology	Limitation	Accuracy
GPS	Poor indoor performance	10 m
Wi-Fi	Density deployment	1 m
Bluetooth	Short range	1 m
Zigbee	Low cost	1 m
RFID	Density deployment	1 m
Infrared	Short range	1 m
Cellular	High cost	20 m
UWB	High cost	0.1 m
Ultrasonic	Short range	0.03 m

**Table 2 sensors-19-03290-t002:** Bit encoding.

**Data bit Encoding**	
Data bit	Sampling bit
1	10
0	11
Synchronization	00
**Sampling Encoding**	
Sampling bit	Frequency
0	2 kHz
1	4 kHz

**Table 3 sensors-19-03290-t003:** Configuration of experimental implementation.

Configuration Parameter	Value
Image sensor architecture	Rolling shutter
Shutter speed	8 kHz
ISO	100
Frame rate	20
Resolution	600 × 800 pixel
f1	2 kHz
f2	4 kHz
Synchronization	Line coding
Stripe with for 4 kHz of f1	3 pixel
Stripe width for 2 kHz of f2	6 pixel
Image processing library	OpenCV
Camera API	Android Camera 2 API (Nexus 5, Google Pixel)
LED size	17 cm
LED power	15 w

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
