# Peer review of "Photography Trilateration Indoor Localization with Image Sensor Communication"

_sensors, 2019, doi:10.3390/s19153290_

Round 1
Reviewer 1 Report
This paper proposes and VLP system based on OCC. This is not a novel idea, and other works have been presented before. It is not clear if any innovation in this issue is proposed in the paper. Therefore, the first recommendation would be that authors explain clearly this point. The paper proposes a novel VLP scheme? Or it only presents experimental results of an implemented system. Other comments are the followings:
Line 230 authors wrote “using the pinhole camera”. I think that the correct sentence would be “using the pinhole camera approximation.”
Line 290 authors introduce “the process includes two main calculations: the image sensor rotation distance and 290 the center shift distance…”, but the explanation is not clear enough. Please extend the explanation properly to understand those concepts.
Several Equations present some parameters or variables not explained. Please detail the meaning of the Equations’ elements in all the cases.
Line 295 figure 4 does not correspond to the following explanation. Please rewrite the paragraph or/and redo the figure 4.
Line 337 Please explain the meaning of x-word and y-word coordinates.
Figure 6 It seems that there is a problem with image references in the figure. For example, point B has its correspondence in the camera image with IA instead of IB. Please check the figure and introduce a better explanation in the figure’s caption.
From line 394 and Equation 16 author introduces the main elements for the coordinates estimations. Some of the parameter needed are the distances from anchor leds to the camera. However, no equation or definitions about how to obtain those distances are presented. Please, explain how the system obtains those distances for calculations.
Line 406 Authors wrote “due to the Gaussian distributions inaccurate of anchor positions…”. Please explain this assumption.
Line 729 Please define CDF properly.
Author Response
First of all, we are grateful for the reviewer’s consideration of our work. All the comments are very useful for us to update and enhance the quality of our manuscript. All the comments we received on this study have been taken into account in improving the quality of the article, and we present our reply to each of them separately as follows. We hope the replies are detailed enough to explain our research.
There are some Figures in our explanation for the reply. So we attached the file to the review system. We hope the reviewer can get it.
Thanks

Reviewer 2 Report
Please see the attachment.

Author Response
First of all, we are grateful for the consideration of the review of our work. All the comments are very useful for us to update and enhance the quality of our manuscript. All the comments we received on this study have been taken into account in improving the quality of the article, and we present our reply to each of them separately as follows. We hope the replies are detailed enough to explain our research.
There are some Figures in the reply document. So we attached the reply document to the review system. We hope that the reviewer can get it.

Round 2
Reviewer 1 Report
Thank you very much for your answers to my comments. They have cleared the paper's contents and increased its quality.
Author Response
We would like to thank the reviewer for a careful and thorough reading of this manuscript and for the thoughtful comments and constructive suggestions, which help to improve the quality of this manuscript.
Reviewer 2 Report
1. The revised version has also responded to the concerns from my previous concerns.
2. The distance estimation between the LED and the lens plane is estimated using Equation (6), where the focal length information should be known in advance. However, the focal length information of the built-in camera module might not be available or may be different in different smartphones. Please comment on its influence on the proposed positioning system.
3. There are still minor issues in the manuscript. The authors are encouraged to improve presentation quality.
a) In line 297, “… form …” should be “… from …”.
b) In the 4th line in Equation (6), the AF2 in the right side should be AD2.
c) The notation h was used in Equation (6) and Equation (20) to denote different terms.
d) The distance estimation error performance was only presented in Figure 29. There is no discussion on the effect of the image sensor rotation, i.e., the pitch and roll, to distance estimation error.
Author Response
We would like to thank the reviewer for a careful and thorough reading of this manuscript and for the thoughtful comments and constructive suggestions, which help to improve the quality of this manuscript. We attach the response in the attachment.
